# Determinants of knowledge, attitude and self-efficacy towards complementary feeding among rural mothers: Baseline data of a cluster-randomized control trial in South West Ethiopia

**Abraham Tamirat Gizaw**[1]*, **Pradeep Sopory**[2], **Morankar Sudhakar**[1]

**1** Faculty of Public Health, Department of Health, Behavior and Society, Institute of Health, Jimma University, Jimma, Ethiopia, **2** Department of Communication, Wayne State University, Detroit, Michigan, United States of America

* abrishntamirat@gmail.com, abraham.tamirat@ju.edu.et

**Data Availability Statement:** All relevant data are within the paper and its Supporting Information files.

## Abstract

### Background

Complementary feeding (CF) is the period when exclusive breastfeeding ends and the introduction of a wide range of foods while breastfeeding should continue until the child is at least 24 months of age. Sub-optimal complementary feeding practices of infants and young children persist due to different factors, which include knowledge, attitude, and self-efficacy of index mothers. Therefore, this study aimed to assess determinants of knowledge, attitude, and self-efficacy towards complementary feeding among rural mothers with index child in rural Ethiopia.

### Methods

A community-based, cross-sectional study was conducted using multistage sampling techniques followed by systematic random sampling techniques. A structured interviewer-administered questionnaire was used. The Chi-square and Fisher's exact probability tests were used to assess the baseline differences in the CF knowledge, attitude, self-efficacy and socio-demographic characteristics of the intervention and control groups. An independent sample t-test was used to determine the mean differences. Multiple linear regression models were fitted to assess the predictors of complementary feeding knowledge, attitude, and self-efficacy. All tests were two-tailed, and a statistically significant association was considered at a $p$-value $\leq 0.05$.

### Results

Overall, 516 mothers were interviewed. 52.5% of the mothers had high complementary feeding (CF) knowledge, whereas only 47.7% and 38.9% had favorable attitude and high self-efficacy, respectively. The socio-demographic characteristics of the intervention and

**Funding:** The authors received no specific funding for this work.

**Competing interests:** The authors have declared that no competing interests exist.

control groups were overall similar. However, there was a significant difference in the child's sex (p = 0.021) and age (p = 0.002). Independent t-tests found no significant difference between the two groups in terms of the mean score of CF knowledge, attitude, and self-efficacy at baseline. Maternal educational status (p = 0.0001), number of ANC visits (p = 0.025), and CF information received (p = 0.011) were significant predictors of CF knowledge. Child sex (p = 0.021) and the number of ANC visits (p = 0.01) were significant predictors of CF attitude. Family size (p = 0.008) and household food security status (p = 0.005) were significant predictors of maternal CF self-efficacy.

## Conclusion

Overall, half of the mothers had high knowledge. Whereas maternal attitudes and self-efficacy toward CF were low. Maternal educational status, the number of ANC visits, and the CF information received were predictors of CF knowledge. Likewise, child sex and the number of ANC visits were predictors of CF attitude. Family size and household food security status were predictors of CF self-efficacy. These findings imply that nutrition intervention strategies are mandatory, particularly to enhance maternal knowledge, attitude, and self-efficacy towards optimum complementary feeding.

## Introduction

Globally, 60% of the 10.9 million child mortalities per year are directly or indirectly attributable to malnutrition. Complementary feeding (CF) which is starting from the age of six months with continued breastfeeding up to two years of age or beyond and it is way to reduce child malnutrition [1]. CF is required in appropriate quantity, quality, and frequency to fulfill children's daily energy needs for growth and development [2].

Notably mothers' poor knowledge and unfavorable attitude towards appropriate foods and feeding practices is often a determinant of inappropriate infant and young child feeding (IYCF) practices than the lack of food [3]. The factors that may contribute to poor CF have been the subject of several research. For instance, a study conducted in Nepal and West Bengal, India indicated that maternal employment, education, age, income and media exposure as determinants of suboptimal CF [4,5]. Whereas inadequate knowledge, illiteracy, and poverty were the main causes of suboptimal CF in rural Bangladesh [6], mother self-efficacy in Iran [7] and knowledge in Ghana [8], were the main causes of suboptimal CF.

There have been different nutritional interventions conducted in Ethiopia in the past decades. For instance, Ethiopia developed and implemented the IYCF guideline in 2004 to improve feeding practice. Messages were given at health institution and community level. However, the majority of mothers inappropriately fed their children [9]. Ethiopia is still in the vicious of inappropriate IYCF practices due to different determinants. The percentage of children stunted and underweight is higher in rural areas than in urban areas, indicating inappropriate child-caring practices in rural areas. According to the study conducted in Jimma, rural Ethiopia showed that 32.3% of children had poor child feeding practices [10]. Another's study conducted in Wollega, Ethiopia revealed that mothers' education status determinant to initiate CF [11]. Similar study conducted in Shashemene showed that the mean age for the introduction of solid, semi-solid and soft foods was 5.6 (SD ± 0.9) months [12]. The finding from Benishangul Gumuz revealed that 22.5% mothers introduced CF before sixth month [13].

Hence, the main aim of this study is to assess determinants of complementary feeding knowledge, attitude, self-efficacy among rural mothers in Maji Woreda, South West region.

## Method

### Study setting and period

This study was carried out in Maji woreda, West Omo zone. The data was collected from March 1 to April 3, 2022. The details about the study setting and period have been described in a study published elsewhere [14].

### Study design and population

Community based cross-sectional study design was employed as baseline study for a cluster-randomized control trial. The details about the study population have been described in a study published elsewhere [14].

### Sample size determination

The sample size was calculated using statcalc (STATA software version 14) with the following assumptions: to detect an increase in appropriate feeding from 7% to 14% [15], with 95% CIs and 80% power, assuming an intra-class correlation coefficient of 0.03 [16]. A total sample size was 516. The details about the study population have been described in a study published elsewhere [14].

### Sample

Multi-stage sampling techniques were used to select mothers who had an infant and a young child aged less than 24 months. Family registration books were used to find mothers. An Excel sheet was formed from the logbook and the households were selected using simple random sampling techniques. Finally, all selected mothers were invited to a meeting at their nearby health posts and farmer training centers, and 100% of the selected mothers attended the meeting. The details about the sampling procedures have been described in a study published elsewhere [14].

### Data collection and measurements

A standardized pretested structured interviewer administered questionnaire was employed for the data collection. The adapted questionnaire was from WHO IYCF indicator parameters [17] previous studies conducted by different researchers. Complementary feeding knowledge questionnaire was adapted after reviewing different studies conducted in different settings [10,18,19]. Components of complementary feeding knowledge assessed were: duration of continued breastfeeding, age of start of complementary feeding, and reasons for giving complementary feeds. The knowledge scale had 10 items and consist of both open- ended and multiple choice questions. Each question was scored 1 for correct and 0 for incorrect answer. The scores summed and a mean score for knowledge questions was computed and respondents who scored less than the mean were labeled as having "low" knowledge and those scored equal to or above the mean were considered as having "high" knowledge.

The attitude scale had 8 items on a 5-point Likert scale. The scale assessed perceived benefits and barriers toward complementary feeding. In order to ensure higher scores denoting positive attitude, negatively worded items were inversely coded during analysis (i.e., 1 = 5, 2 = 4, 3 = 3, 4 = 2, 5 = 1). Total scores were generated for each participant [20,21]. A mean score for attitude questions was computed and respondents who scored below the mean were

considered as having an "unfavorable" attitude and those scored equal to or above the mean were considered as having a "favorable "attitude.

Self-efficacy consisting of 9-items with a five-point Likert scale, developed to measure mothers' confidence towards CF adapted from previous study [22]. Components of the self-efficacy assessed were: confidence in giving a variety of meals and feeding frequency. A mean score for self-efficacy questions was computed and respondents who scored below the mean were considered as having an "low" self-efficacy and those scored equal to or above the mean were considered as having a "high "self-efficacy.

The birth weight of infants and young children was obtained from the registration log for those only who gave birth at the health facilities. However, for that mother who gave birth at home, there is no weight measurement included.

## Data quality control

The questionnaire was prepared in English and then translated to Amharic and then back translated to English by expert of the language to keep its consistency. Pretesting of the questionnaire was done on 5% in the Bench-Sheko zone (other than the study area). *Cronbach's alpha* was determined (it was 0.88; the acceptable range is >0.7) to assess the reliability of the questionnaire before the actual data collection. Daily supervision was conducted by the supervisors and researcher before the study period and appropriate changes were made in the questionnaire.

## Data analysis

Double data entries were done using EpiData (version 3.1), and all statistical analyses were conducted using SPSS version 23. The data were summarized using frequencies and percentages. The Chi-square and Fisher's exact probability tests were used to assess the baseline differences in the socio-demographic characteristics of the two groups. An independent sample t-test was used to determine the mean differences in complementary feeding knowledge, attitude, and self-efficacy. Multiple linear regression models were fitted to assess the predictors of CF knowledge, attitude, and self-efficacy. The scores for the three predictors were standardized based on the distribution of the data, and the results are expressed as regression coefficients with 95% confidence intervals (95%CIs). All tests were two-tailed and a statistically significant association was considered at a $p$-value $\leq 0.05$.

## Ethical approval and consent to participate

The study received ethical approval (reference number. IHRPG/938/20) from Jimma University's Institute of Health Research and Postgraduate Office, Institutional Review board. Administrative permission was acquired from Maji Woreda Administrative offices, and formal letters to the research area were obtained from Maji Woreda Health Office. All participants provided written informed consent. Participation was voluntary with the right to withdraw at any time.

## Results

### Baseline socio-demographic characteristics of participants

As presented in Table 1, except for child sex and child age, baseline infant and young child, maternal, and household characteristics were comparable between the intervention and control groups. The details about baseline socio-demographic characteristics of participants have been described in a study published elsewhere [14].

**Table 1. Socio-demographic characteristics of mothers, in Southwest Ethiopia.**

| Variables | Intervention (n = 258) | | Control (n = 258) | | X$^2$ test | p-value |
|---|---|---|---|---|---|---|
| | n | (%) | n | (%) | | |
| **Mother's age (in years)** | | | | | | |
| 18–24 | 38 | 14.23 | 38 | 14.23 | 0.07 | 0.967 |
| 25–34 | 130 | 51.55 | 136 | 51.55 | | |
| 35–49 | 90 | 34.22 | 84 | 34.22 | | |
| M±SD | 31.69 ± 7.74 | | 30.83 ± 7.01 | | | |
| **Marital status** | | | | | | |
| Married | 247 | 95.74 | 251 | 97.30 | 0.92 | 0.337 |
| Divorced | 11 | 4.26 | 7 | 2.70 | | |
| **Religion** | | | | | | |
| Orthodox Christian | 166 | 64.34 | 171 | 66.28 | 0.21 | 0.644 |
| Protestant | 92 | 35.66 | 87 | 33.72 | | |
| Maternal occupation[†] | | | | | | |
| Housewife/farmer | 256 | 99.22 | 255 | 98.84 | | 0.999 |
| Government employee | 2 | 0.78 | 3 | 1.16 | | |
| **Family size** | | | | | | |
| 1–3 | 58 | 22.48 | 58 | 22.48 | 3.02 | 0.221 |
| 4–6 | 131 | 50.77 | 147 | 56.98 | | |
| ≥7 | 69 | 26.75 | 53 | 20.54 | | |
| **Monthly income of the household (ETB)** | | | | | | |
| ≤500 | 140 | 54.26 | 124 | 48.06 | 2.53 | 0.639 |
| 500–1000 | 90 | 34.88 | 98 | 37.98 | | |
| 1000–1500 | 14 | 5.43 | 16 | 6.20 | | |
| 1501–2000 | 8 | 3.10 | 12 | 4.65 | | |
| ≥2000 | 6 | 2.33 | 8 | 3.11 | | |
| **Maternal educational status** | | | | | | |
| Illiterate | 97 | 37.58 | 91 | 35.27 | 0.30. | 0.859 |
| Primary school | 141 | 54.65 | 146 | 56.59 | | |
| Secondary school and higher | 20 | 7.77 | 21 | 8.14 | | |
| **Household food security status** | | | | | | |
| Secured | 14 | 5.43 | 13 | 5.04 | 0.04 | 0.843 |
| Not secured | 244 | 94.57 | 245 | 94.96 | | |
| **Child sex** | | | | | | |
| Male | 157 | 60.85 | 131 | 50.77 | 5.31 | 0.021* |
| Female | 101 | 39.15 | 127 | 49.23 | | |
| **Child age (months)** | | | | | | |
| 0–5 | 41 | 15.89 | 45 | 17.44 | 15.03 | 0.002* |
| 6–11 | 85 | 32.94 | 85 | 32.94 | | |
| 12–17 | 118 | 45.74 | 90 | 34.88 | | |
| 18–24 | 14 | 5.43 | 38 | 14.74 | | |
| M±SD | **10.97 ± 4.96** | | **11.39 ± 5.80** | | | |
| **Birth order** | | | | | | |
| 1[st] | 45 | 17.44 | 44 | 17.05 | 0.09 | 0.958 |
| 2[nd] - 4[th] | 171 | 66.28 | 174 | 67.44 | | |
| 5[th] or more | 42 | 16.28 | 40 | 15.51 | | |
| **Number of ANC visits** | | | | | | |

*(Continued)*

**Table 1.** (Continued)

| Variables | Intervention (n = 258) | | Control (n = 258) | | $X^2$ test | p-value |
|---|---|---|---|---|---|---|
| | n | (%) | n | (%) | | |
| No ANC visits | 118 | 45.74 | 116 | 44.96 | 0.04 | 0.981 |
| <4 visits | 112 | 43.41 | 113 | 43.8 | | |
| ≥4 visits | 28 | 10.85 | 29 | 11.24 | | |
| **Received complementary feeding information** | | | | | | |
| No | 189 | 73.25 | 188 | 72.87 | 0.39 | 0.731 |
| Yes | 69 | 26.75 | 70 | 27.13 | | |
| **Place of delivery** | | | | | | |
| Home | 173 | 67.05 | 184 | 71.32 | 1.1 | 0.294 |
| Health institution | 85 | 32.95 | 74 | 28.68 | | |
| Delivery type[†] | | | | | | |
| Normal vaginal delivery | 254 | 98.45 | 255 | 98.84 | | 0.999 |
| Caesarian section | 4 | 1.55 | 3 | 1.16 | | |
| **Postnatal care** | | | | | | |
| No | 214 | 82.94 | 225 | 87.21 | 1.85 | 0.174 |
| Yes | 44 | 17.06 | 33 | 12.79 | | |
| **Number of children** | | | | | | |
| 1–2 | 50 | 19.38 | 58 | 22.48 | 2.11 | 0.349 |
| 3–4 | 130 | 50.39 | 136 | 52.71 | | |
| ≥5 | 78 | 30.23 | 64 | 24.81 | | |
| **Parity** | | | | | | |
| Primiparous | 45 | 17.44 | 44 | 17.05 | 0.01 | 0.907 |
| Multiparous | 213 | 82.56 | 214 | 82.95 | | |

Chi$^2$ test

*significant at p < 0.05; [†]Fisher's exact probability test.

**Baseline CF knowledge, attitude and self-efficacy among the groups.** There were no statistically significant differences between the intervention and control groups in terms of the outcome variables at the baseline (Table 2).

## Predictors of CF knowledge, attitude, and self-efficacy

A multivariate analysis was used to test the effect of socio-demographic, obstetric, and healthcare variables on knowledge, attitude, and self-efficacy regarding complementary feeding. The results are presented in Table 3. Maternal educational status (β = 0.49, p = 0.0001), number of

**Table 2. Baseline comparison of the mean total CF knowledge, attitude and self-efficacy scores of the intervention and control groups.**

| Breastfeeding variables | Mean ± SD | | t-test | Mean difference (95%CI) | P-value |
|---|---|---|---|---|---|
| | Intervention (n = 258) | Control (n = 258) | | | |
| CF Knowledge score | 3.98±0.34 | 4.06±0.31 | 0.045 | 0.09 (-1.14, 1.03) | 0.922 |
| CF Attitude score | 21.41±2.01 | 20.89±2.71 | 0.025 | 1.65 (1.04,2.11) | 0.871 |
| CF Self-efficacy score | 19.93±1.06 | 20.02±1.01 | -0.018 | -0.29 (-1.18,1.02) | 0.906 |

**Table 3. Predictors of knowledge, attitude and self-efficacy towards complementary feeding among mothers.**

| Variable | Knowledge | | | Attitude | | | Self-efficacy | | |
|---|---|---|---|---|---|---|---|---|---|
| | β | SE | p-Value | β | SE | p-Value | β | SE | p-Value |
| Mother's age | 0.51 | 1.25 | 0.153 | 0.62 | 1.52 | 0.231 | 0.91 | 1.78 | 0.451 |
| Marital status | 0.99 | 1.89 | 0.741 | -0.19 | 2.89 | 0.413 | 0.50 | 1.09 | 0.442 |
| Maternal occupation | 0.33 | 1952.75 | 0.153 | 0.56 | 2.58 | 0.147 | 0.43 | 1.87 | 0.216 |
| Family size | 0.84 | 2.85 | 0.542 | 0.59 | 1.52 | 0.289 | -0.28 | 0.987 | 0.008* |
| Monthly income | 0.88 | 1.85 | 0.287 | 0.51 | 0.19 | 0.404 | 0.31 | 1.04 | 0.093 |
| Maternal educational status | 0.49 | 2.01 | 0.0001* | 0.74 | 3.01 | 0.485 | 0.63 | 3.02 | 0.191 |
| Household food security status | 0.35 | 1.57 | 0.223 | -0.33 | 1.88 | 0.856 | -1.49 | 1.06 | 0.005* |
| Child sex | 0.58 | 2.01 | 0.155 | 1.71 | 2421.85 | 0.021* | 0.81 | 3.01 | 0.318 |
| Child age | 0.99 | 1.56 | 0.128 | 0.29 | 2.99 | 0.101 | 0.51 | 4.22 | 0.143 |
| Number of ANC visits | 0.91 | 2.61 | 0.025* | 0.78 | 4.92 | 0.001 * | 0.66 | 2.88 | 0.352 |
| Received CF information | 0.65 | 2.88) | 0.011* | 0.88 | 3.57 | 0.221 | 0.44 | 4.33 | 0.317 |
| Birth weight | 1.45 | 3.33 | 0.98 | 0.85 | 1.88 | 0.65 | 0.91 | 1.96 | 0.658 |
| Postnatal care | 0.85 | 1.43 | 0.108 | 0.99 | 3.99 | 0.442 | 0.59 | 2.88 | 0.148 |
| Parity | 0.66 | 2.19 | 0.124 | 0.22 | 3.47 | 0.138 | -0.21 | 1.23 | 0.991 |
| Constant | -11.69 | 1287.33 | 0.864 | -6.81 | 1677.19 | 0.991 | 1.09 | 1.04 | 0.421 |

β: *Regression coefficients*. SE: *Standard error*.

*p<0.05, significant values.95% confidence intervals. Abbreviations: ANC, antenatal care.

ANC visits (β = 0.91, p = 0.025) and CF information received (β = 0.65, p = 0.011) were significant predictors of CF knowledge. An increase in maternal educational level, an increase in the number of ANC visits, and receiving CF information by one-unit increase CF knowledge by a factor of 0.49, 0.91, and 0.65, respectively.

Regarding CF attitude, child sex (β = 1.71, p = 0.021) and the number of ANC visits (0.78, p = 0.01) were significant predictors. Having a female child is associated with a 1.71-fold increase in attitude and an increase in the number of ANC visits in one unit, which increases maternal CF attitude by 0.78. Family size (β = -0.28, p = 0.008) and household food security status (β = -1.49, p = 0.005) were significant predictors of maternal CF self-efficacy. On the other hand, as the family size increases within one unit, self-efficacy is reduced by a factor of 0.28 and living in a food-insecure household is associated with a 1.49-fold decrease in the level of self-efficacy.

## Discussion

In this study, we investigated the level and determinants of maternal knowledge, attitude and self-efficacy toward complementary feeding (CF). Even though Ethiopia has attempted to improve complementary feeding, its success has been limited. To the best of our knowledge, there are only a few studies that have used validated questionnaires to assess complementary feeding knowledge and attitude in Ethiopia. However, there is no single study conducted to assess complementary feeding self-efficacy among mothers. Therefore, the baseline survey provides a novel insight into the study of complementary feeding self-efficacy among mothers in addition to knowledge and attitude. Hence, the evidence generated from this study is particularly important for planning appropriate nutrition interventions that can potentially improve child feeding practices and, thus, children's growth and survival.

The results of this study suggest that maternal educational status, number of antenatal care visits, and receiving complementary feeding information were significant predictors of knowledge about complementary feeding. Maternal educational level, number of antenatal care

visits, and having a female child were found to be significant predictors of attitude towards complementary feeding. Additionally, family size and household food security status were significant predictors of maternal self-efficacy regarding complementary feeding. These findings highlight the importance of maternal education, healthcare utilization, and household factors in influencing knowledge, attitude, and self-efficacy towards complementary feeding.

The pertinent finding of this study was that more than half (52.5%) of the mothers had high knowledge about CF. Our finding is similar to the (52.0%) mothers with high knowledge reported in Ghana [20]. However, the finding of this study is lower when compared with the study conducted in North West Ethiopia, where (60.0%) of the mothers had good knowledge about CF [18]. This difference may be explained by the fact that there are socio-demographic differences in the study areas and due to access to information related to complementary feeding.

An important finding of this study was that an increase in maternal educational status is associated with an increase in complementary feeding knowledge. This may be explained by the fact that literate mothers can understand the nutrition message disseminated by health professionals or other sources of information. This study is in line with the studies conducted in northwestern Ethiopia [18], Nigeria [23], and Poland [24]. Similarly, mothers who had four or more (≥4 visits) ANC visits had high knowledge. This may be explained by the fact that during these visits, mothers are educated on proper nutrition, including the introduction of complementary foods at six months of age. This education helps the mother make informed choices about what to feed their child and how much to feed them. On the other hand, the effectiveness of ANC visits increases when multiple visits are carried out, as it allows healthcare providers to promote health behaviors related to children and mothers. This study is also in congruence with previous studies conducted in Ethiopia [18,25–27].

Maternal attitude is the most important psychometric variable that predicts optimum complementary feeding. This study revealed that below half (47.7%) of the mothers had a favorable attitude towards complementary feeding. This result was lower than the study conducted in northwestern Ethiopia [18] and Assosa [28]. A number of reasons may be given for these differences, topmost among them was the previous study conducted in a semi-urban setting so that the participating mothers had exposure to appropriate feeding information from different sources.

This study also revealed that maternal attitude is associated with child sex. The finding of this study is in congruence with a study conducted in Rwanda [29], where favorable attitude was associated with child sex. However, contrary to the previous studies conducted in Ethiopia [18] and Uganda [30]. Furthermore, the number of ANC visits was significantly associated with maternal attitude. This study is in congruence with the studies conducted in Ethiopia [18,31] and Ghana [32]. This is due to the fact that when there is exposure to information, mothers will add their own feelings and form an attitude.

Maternal self-efficacy toward complementary feeding is another important psychometric variable that this study tried to investigate. The findings of this study further affirm that the mothers in the study area had low self-efficacy (61.1%). The finding of this study is much lower than the study conducted in Bangladesh [22]. A number of reasons may be given to this variation in self-efficacy; topmost among them was household food security, which was better in Bangladesh than Ethiopia and particularly in the study area.

Another important finding of this study was that as family size increased within one unit, self-efficacy decreased by a factor of 0.28. This is because as the number of families in resource limited setting increases, it imposes difficulties to maintain the recommended optimum complementary feeding. This finding is in congruence with a studies done in Ghana [8], in Senegal [33] and Uganda [34]. Finally, living in a food-insecure household was associated with a

1.49-fold decrease in the level of self-efficacy. This is due to the fact that the unavailability of different food items may hinder the mothers' ability to maintain the desired frequency and consistency of the recommended optimum complementary feeding. The finding of this study is in congruence with previous studies conducted in Ethiopia [35] and Bangladesh [36].

## Strength and limitation of the study

The questionnaire used for this study is based on WHO IYCF indicator parameters. A standard pretested questionnaire was used. This paper is the first to be conducted in the study area and complementary feeding self-efficacy was not conducted elsewhere in Ethiopia. However, the limitation of this study might have had an information bias for the knowledge, attitude, and self-efficacy questions, which might have led to the over-reporting of the desirable answers due to a fear of being judged. Second, since it was a cross-sectional study, causal inferences between variables cannot be investigated.

## Conclusion

Overall, half of the mothers had high knowledge about complementary feeding. Whereas maternal attitudes and self-efficacy toward complementary feeding were low. Maternal educational status, information received, and number of ANC visits were predictors of a high maternal knowledge score. Likewise, child sex and the number of ANC visits were predictors of the maternal attitude score. Finally, family size and household food insecurity status were predictors of maternal self-efficacy. These findings imply that nutrition intervention strategies are mandatory, particularly to enhance maternal knowledge, attitude, and self-efficacy towards optimum complementary feeding.

## Supporting information

**S1 Checklist. STROBE statement—checklist of items that should be included in reports of observational studies.**
(DOCX)

**S1 Table. Descriptive statistics of CF knowledge, attitude and self-efficacy.**
(DOCX)

## Acknowledgments

We would like to thank Jimma University, *West Omo* Zone Health Department office for facilitating the data collection process, health extension workers, and the women development army working in the kebele. Finally, we would like to thank the mothers who participated in the study.

## Author Contributions

**Conceptualization:** Abraham Tamirat Gizaw, Pradeep Sopory, Morankar Sudhakar.

**Data curation:** Abraham Tamirat Gizaw.

**Formal analysis:** Abraham Tamirat Gizaw.

**Investigation:** Abraham Tamirat Gizaw, Pradeep Sopory, Morankar Sudhakar.

**Methodology:** Abraham Tamirat Gizaw, Pradeep Sopory, Morankar Sudhakar.

**Project administration:** Abraham Tamirat Gizaw.

**Software:** Abraham Tamirat Gizaw, Pradeep Sopory.

**Supervision:** Pradeep Sopory, Morankar Sudhakar.

**Validation:** Pradeep Sopory, Morankar Sudhakar.

**Visualization:** Morankar Sudhakar.

**Writing – original draft:** Abraham Tamirat Gizaw.

**Writing – review & editing:** Abraham Tamirat Gizaw, Pradeep Sopory, Morankar Sudhakar.

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
