## [Decision Letter · Decision Letter 0]

31 Jan 2023

PONE-D-22-26621Determinants of knowledge, attitude and self-efficacy towards complementary feeding among rural mothers: Baseline data of a cluster-randomized control trial in South West Ethiopia

PLOS ONE

Dear Dr. Gizaw,

Thank you for submitting your manuscript to PLOS ONE. After careful consideration, we feel that it has merit but does not fully meet PLOS ONE’s publication criteria as it currently stands. Therefore, we invite you to submit a revised version of the manuscript that addresses the points raised during the review process.

*During internal evaluation of your mansucript, we have noted that you have submitted highly similar article which is currently under consideration entitle ‘’Breastfeeding knowledge, attitude, and self-efficacy among mothers with infant and young child (0-24 months): Baseline data of a cluster-randomized control trial in South West Ethiopia’’ *

*In addition to the reviewer’s comments, we would require that you cite and discuss the above mentioned related paper in the body of the manuscript (introduction, methods if overlap in participants and discussion). In particular, please discuss how the current mansucript contributes to the basis of academic knowledge in light of the above-mentioned mansucript. Please bear in mind that our publication criteria state " If a submitted study replicates or is very similar to previous work, authors must provide a sound scientific rationale for the submitted work and clearly reference and discuss the existing literature. Submissions that replicate or are derivative of existing work will likely be rejected if authors do not provide adequate justification.*

We look forward to receiving your revised manuscript.

Kind regards,

Lucinda Shen

Staff Editor 

on behalf of 

Mohan Kumar

Academic Editor

PLOS ONE

Journal Requirements: 

Reviewers' comments:

Reviewer's Responses to Questions

**Comments to the Author**

1. Is the manuscript technically sound, and do the data support the conclusions?

Reviewer #1: Yes

Reviewer #2: Yes

2. Has the statistical analysis been performed appropriately and rigorously? 

Reviewer #1: Yes

Reviewer #2: Yes

3. Have the authors made all data underlying the findings in their manuscript fully available?

Reviewer #1: No

Reviewer #2: Yes

4. Is the manuscript presented in an intelligible fashion and written in standard English?

Reviewer #1: Yes

Reviewer #2: Yes

5. Review Comments to the Author

Reviewer #1: The authors have made a very good effort overall to describe their findings. However, some areas they further need to address.

Abstract:

Keywords: pls keep it in alphabetical order.

Introduction is overall well written, and although as the authors stated there are

poor knowledge about appropriate foods and feeding practices is often a determinant of inappropriate infant and young child feeding (IYCF) practices, can the authors provide results on attitude on complementary feeding from middle and/or low income countries?

Methods:

Sample

Sufficient details were not provided on the sampling approach for the recruitment of participants. Please provide sufficient detail.

Authors stated “All selected mothers were invited to a meeting in their nearby health posts and farmer training centers”. Pls mention response rate.

Data collection and measurements

This paragraph seems confusion to readers “Components of the attitude towards complementary feeding consisted of 8 items with a five-point Likert scale, rating maternal attitude towards complementary feeding translated from English. that assessed: giving a variety of meals and feeding frequency”. Pls re-write.

Components of complementary feeding knowledge, attitude and self- efficacy described well, however, tools as complementary file may helpful. Who involved in data collection? How confidentiality were ascertained as authors mention “All selected mothers were invited to a meeting in their nearby health posts and farmer training centers”.

How birth weight are recorded. From hospital records or anything else..??

Results

Table 1: Child sex-Bold.

Pls mention whether responses for “Sources of information” are multiple response or mutually exclusive. Omit cell with zero values.

Results are difficult to read and follow (especially the regression analyses) and overall, not explained correctly. Also, grammatical errors are present.

Discussion: although brief, the discussion adequately compares and explains the results in relation to other studies. The notion of collective measures as an intervention is appropriate but further explanation/ description would be helpful, particularly for readers working in similar environments. Furthermore, it would be useful to know about the efficacy of such interventions in other parts of Ethiopia or other countries such as countries.

Reviewer #2: This article has investigated the level and determinants of maternal knowledge, attitude and self efficacy toward complementary feeding.

- definition of complementary feeding had not been described in the abstract.

- In the discussion, this sentence needs a revision – “This study also revealed that factors that determines complementary feeding knowledge. Mother who had secondary school and above had higher mean knowledge score. This may be explained by the fact that mothers with formal educational background can understand the nutrition message disseminated by health professionals or other sources of information.” – In fact, the study did not reveal that mothers who had secondary school are more likely to understand the nutrition message. In the discussion it would be interesting to learn how to approach the most vulnerable mothers to understand the message.

- “Similarly, mothers who had four and more (=4 visits) ANC visits were high knowledge score. This may be explained with the fact mothers exposed to the nutrition education and counselling during their visits.” – this information should be discussed in further detail, for instance explaining how was the ANC visit and describe the reasons for effectiveness when four or more visits had been carried out.

- “This finding is below study conducted in North West Ethiopia (51%) [18] and Assosa, Ethiopia [28].” – this sentence needs to be reviewed.

- “However, the limitation of the study is that the finding of the study will not be generalizable to the urban setting.” – it should be described the limitations o the design study as a community-based cross-sectional study – such as reverse causality, misclassification bias, selection bias and confounding.

6. PLOS authors have the option to publish the peer review history of their article (what does this mean?). If published, this will include your full peer review and any attached files.

Reviewer #1: **Yes: **Jitendra K. Singh

Reviewer #2: **Yes: **Airton Tetelbom Stein

---

## [Author Response · Author response to Decision Letter 0]

16 Mar 2023

To: Dr. Richard Ibañez Dilla (PLoS ONE) 

Manuscript number: PONE-D-22-26621 

Manuscript title: “Determinants of knowledge, attitude and self-efficacy toward complementary feeding among rural mothers: Baseline data of a cluster-randomized control trial in South West Ethiopia”

Comment 

1. Please include a copy of Table 2 which you refer to in your text on page 10.

Response 

Corrected and the copy of Table 2 included. It was a mislabeling of the table and the number is corrected. 

Comment 

Response 

We declare that we haven’t received any funding from funders. Therefore “The authors received no specific funding for this work.”

---

## [Decision Letter · Decision Letter 1]

6 Jul 2023

PONE-D-22-26621R1Determinants of knowledge, attitude and self-efficacy towards complementary feeding among rural mothers: Baseline data of a cluster-randomized control trial in South West EthiopiaPLOS ONE

Dear Dr. Gizaw,

Thank you for submitting your manuscript to PLOS ONE. After careful consideration, we feel that it has merit but does not fully meet PLOS ONE’s publication criteria as it currently stands. Therefore, we invite you to submit a revised version of the manuscript that addresses the points raised during the review process.

We look forward to receiving your revised manuscript.

Kind regards,

Mohan Kumar

Academic Editor

PLOS ONE

Additional Editor Comments:

Kindly provide a point to point detailed response to reviewer comments, highlighting the modification done, page and line numbers.

Reviewers' comments:

Reviewer's Responses to Questions

**Comments to the Author**

1. If the authors have adequately addressed your comments raised in a previous round of review and you feel that this manuscript is now acceptable for publication, you may indicate that here to bypass the “Comments to the Author” section, enter your conflict of interest statement in the “Confidential to Editor” section, and submit your "Accept" recommendation.

Reviewer #1: All comments have been addressed

Reviewer #2: All comments have been addressed

2. Is the manuscript technically sound, and do the data support the conclusions?

Reviewer #1: Yes

Reviewer #2: Yes

3. Has the statistical analysis been performed appropriately and rigorously? 

Reviewer #1: Yes

Reviewer #2: Yes

4. Have the authors made all data underlying the findings in their manuscript fully available?

Reviewer #1: Yes

Reviewer #2: Yes

5. Is the manuscript presented in an intelligible fashion and written in standard English?

Reviewer #1: Yes

Reviewer #2: Yes

6. Review Comments to the Author

Reviewer #1: (No Response)

Reviewer #2: In the abstract:

It has not been addressed the following topics: How data had been collected?, Which are the variables being compared? It is not described the two groups that had been compared.

This sentence needs to be reviewed: "Except for the child’s sex and age, no significant difference was observed between the intervention and control groups in terms of socio-demographic variables (p > 0.05)." - There is a need to present a confidence interval for the variables that had been identified as statistically significant.

In the discussion there is a need to present the synthesis of the main results found in this study.

7. PLOS authors have the option to publish the peer review history of their article (what does this mean?). If published, this will include your full peer review and any attached files.

Reviewer #1: **Yes: **Dr. Jitendra Kumar Singh

Reviewer #2: **Yes: **Airton Tetelbom Stein - Federal University of Health Sciences of Porto Alegre and Conceicao Hospital of Porto Alegre

---

## [Author Response · Author response to Decision Letter 1]

7 Jul 2023

To: Dr. Mohan Kumar (Academic editor) 

PLoS ONE

Manuscript number : PONE-D-22-26621 

Manuscript title: “Determinants of knowledge, attitude and self-efficacy toward complementary feeding among rural mothers: Baseline data of a cluster-randomized control trial in South West Ethiopia”

We thank the academic editor and reviewers for their comments and appreciate the opportunity to submit a revised manuscript. The manuscript has been revised in accordance with the comments and suggestions. Overall, the comments are appreciation on the manuscript quality and write-up and really we appreciate for your rigor reviews.

Response to editor and reviewer Response Page Number

Response to Reviewer 1

In the abstract:

It has not been addressed the following topics: 

How data had been collected?, -We have accepted the comment and addressed the comment in the method-section of the abstract. Page 2

Which are the variables being compared? -We have accepted the comment and addressed the comment in the method-section of the abstract. Page 2

It is not described the two groups that had been compared.

 -We have accepted the comment and addressed the comment in the method-section of the abstract. Page 2

This sentence needs to be reviewed: "Except for the child’s sex and age, no significant difference was observed between the intervention and control groups in terms of socio-demographic variables (p > 0.05)." 

-We have accepted the comment and addressed (revised and rewrote) the sentence accordingly in the result-section of the abstracts. Page 2

There is a need to present a confidence interval for the variables that had been identified as statistically significant.

 We have accepted the comment and addressed the sentence accordingly in the result-section of the abstracts. Page 2

Discussion 

In the discussion there is a need to present the synthesis of the main results found in this study.

 We have accepted the comment and addressed the comment in the discussion part of the manuscript. Page 13

We thank you the editors and reviewer for the onstructive comments.

---

## [Decision Letter · Decision Letter 2]

10 Oct 2023

Determinants of knowledge, attitude and self-efficacy towards complementary feeding among rural mothers: Baseline data of a cluster-randomized control trial in South West Ethiopia

PONE-D-22-26621R2

Dear Dr. Gizaw,

We’re pleased to inform you that your manuscript has been judged scientifically suitable for publication and will be formally accepted for publication once it meets all outstanding technical requirements.

Kind regards,

Jennifer Tucker, Staff Editor, on behalf of

Abdu Oumar

Academic Editor

PLOS ONE

Additional Editor Comments (optional):

Whilst reviewer 2 has provided additional comments alongside their decision, these are not required revisions for your manuscript.

Reviewers' comments:

Reviewer's Responses to Questions

**Comments to the Author**

1. If the authors have adequately addressed your comments raised in a previous round of review and you feel that this manuscript is now acceptable for publication, you may indicate that here to bypass the “Comments to the Author” section, enter your conflict of interest statement in the “Confidential to Editor” section, and submit your "Accept" recommendation.

Reviewer #2: All comments have been addressed

2. Is the manuscript technically sound, and do the data support the conclusions?

Reviewer #2: Yes

3. Has the statistical analysis been performed appropriately and rigorously? 

Reviewer #2: Yes

4. Have the authors made all data underlying the findings in their manuscript fully available?

Reviewer #2: Yes

5. Is the manuscript presented in an intelligible fashion and written in standard English?

Reviewer #2: Yes

6. Review Comments to the Author

Reviewer #2: There is a need to present the possibility of misclassification bias which was not described in the discussion section.

The first paragraph of discussion should present the main results of the present study.

7. PLOS authors have the option to publish the peer review history of their article (what does this mean?). If published, this will include your full peer review and any attached files.

Reviewer #2: **Yes: **Airton Tetelbom Stein

---

## [Editor Report · Acceptance letter]

12 Oct 2023

PONE-D-22-26621R2 

Determinants of knowledge, attitude and self-efficacy towards complementary feeding among rural mothers: Baseline data of a cluster-randomized control trial in South West Ethiopia 

Dear Dr. Gizaw:

I'm pleased to inform you that your manuscript has been deemed suitable for publication in PLOS ONE. Congratulations! Your manuscript is now with our production department. 

Kind regards, 

on behalf of

Dr. Abdu Oumer 

Academic Editor

PLOS ONE